# Adapted Original Music as an Environmental Enrichment in an Intensive Pig Production System Reduced Aggression in Weaned Pigs during Regrouping

**DOI:** 10.3390/ani13233599

**Published:** 2023-11-21

**Authors:** Natalia Alvarez-Hernandez, Darío Vallejo-Timarán, Berardo de Jesús Rodriguez

**Affiliations:** 1QUIRON Pathobiology Research Group, Faculty of Agricultural Sciences, Ciudadela de Robledo, University of Antioquia—UdeA, Carrera 75 No. 65-87, Medellín 050034, Colombia; berardo.rodriguez@udea.edu.co; 2Colombian Agricultural Research Corporation—AGROSAVIA, Obonuco Research Center, Pasto 520038, Colombia; davallejo@agrosavia.co

**Keywords:** animal welfare, agonistic behavior, music, play behavior, mixing pigs

## Abstract

**Simple Summary:**

In intensive swine production systems, managing the regrouping of pigs is a common practice, but it often leads to aggressive behaviors, which can harm the welfare of the animals. This study explores an approach that involves composing and producing music based on acoustic parameters established by QUIRON Pathobiology research group. The aim is to reduce aggressive behaviors in pigs, thereby enhancing pig welfare during regrouping. Our findings indicate that this cost-effective and easy-to-implement strategy reduces aggressive behaviors in piglets during regrouping. This research offers valuable insights for producers, providing them with a practical way to enhance pig behavior and welfare while also contributing to the broader understanding of animal well-being in swine production systems.

**Abstract:**

In intensive swine production systems,, the practice of regrouping unfamiliar pigs is common, often leading to aggressive behavior. Although the effect of different musical genres composed for humans has been evaluated in pigs to mitigate aggression, there have been few attempts to create music specifically for pigs. Here, we assess whether sensory stimulation through music, created by adapting the acoustic parameters in the sound mix, induces changes in the aggressive behaviors of pigs during regrouping. Six litters of 10-week-old piglets were randomly selected and assigned to different treatments. The control group (Group A) received no intervention, while Group B was exposed to music for two continuous hours in the morning and afternoon from the time of regrouping. Group C received musical stimulation for one continuous hour in the morning following regrouping. A significant reduction in the frequency and duration of aggressive behaviors was observed in the groups that received musical stimulation during regrouping. Additionally, social, and individual play behaviors showed a decrease in the musical stimulation groups. These findings provide evidence for the effectiveness of created music as a strategy in reducing aggressive behavior during pig regrouping, which can enhance the welfare of pigs and offer a practical solution for pig producers to minimize aggression and its associated negative impacts.

## 1. Introduction

In intensive swine production systems, maintaining good husbandry conditions and animal welfare is crucial. However, intensive swine production systems frequently induce stress due to factors like environmental conditions, transportation, and social interactions. Issues such as barren and overcrowded spaces without straw for bedding or rooting contribute to problems such as tail biting, impacting both the economic viability and welfare of pig production [1]. During transportation, concerns like fatigue, heat stress, and aggressive behavior are common welfare issues affecting pigs [2]. These issues not only impact physical health but also present behavioral challenges, as observed during regrouping.

Pigs are gregarious animals that establish hierarchical relationships to determine the order of access to resources such as food, water [3], mates, and rest [4]. Fighting and avoidance behaviors are how the hierarchy is exercised. Under natural conditions, the hierarchy remains more stable, resulting in less threatening and aggressive behavior [5]. However, under commercial conditions, the grouping or mixing of unfamiliar pigs (at weaning, during fattening, or in breeding females) is common, favoring aggressive behaviors among them as a way to establish a hierarchy [6,7,8], impacting the welfare of individual pigs and posing management challenges in intensive swine production systems.

Aggressive or agonistic behaviors consist of offensive movements in which the pig tries to bite mainly the head region, whereas defensive behaviors are those in which the pig avoids being bitten by adopting parallel or reverse parallel postures, turning its body, or tilting its head [5]. The level of aggression depends on several factors such as weight, [7] space, density [9], and the degree of familiarity among individuals [8]. Aggressive encounters can be associated with skin lesions, reduced immune response [10], abortions, and reduced weight gain, [7] which has implications for producer profitability. Economic losses associated to pig carcass confiscation due to abscesses caused by tail lesions were estimated to be €0.37 per animal slaughtered [11]. According to the findings, when tail-biting lesions occur at an estimated frequency of approximately 10%, the related costs arising from this detrimental behavior can rise to €2.3 per pig at slaughter, equivalent to roughly 1.6% of the total carcass value [12]. However, the financial burdens linked to each pig experiencing tail-biting lesions were significantly higher, as simulations indicated a range of €16 to €35 per affected pig. These costs were found to vary depending on the prevalence of tail-biting lesions. It is suggested that the costs of tail-biting lesions across Europe are approximately €2.0 (±€1.4) per finished pig [12].

Strategies such as early socialization during the first two weeks of life, regrouping pigs by sex at weaning [13], the use of synthetic maternal pheromones [14], and the use of tryptophan in the diet [15], have been used to improve social skills in piglets and reduce aggressive interactions in future mixes, as well as the amount of time animals spend fighting. It has been demonstrated that decreasing the available area per individual, thereby increasing group density, heightens the frequency of agonistic interactions within the group. Therefore, increasing space and reducing stock densities may effectively mitigate piglet aggression [16]. However, it is essential to emphasize that a fundamental principle of the weaning process is to avoid excessive random mixing to prevent piglet aggression [17]. Although in intensive production systems, there is limited acceptance of alternatives such as early socialization to reduce aggressive behavior during the regrouping process [18]. This limited acceptance can be explained by the low prioritization of the problem [19], impractical enrichment alternatives [18,19] a lack of information on cost-effectiveness, and ineffective communication between researchers/technicians and the farming community [18,20].

Environmental enrichment has been used as a strategy to control and minimize various stressors to which animals are exposed in production environments. Of these, the use of music is one type of enrichment that could be implemented in production environments to control and minimize environmental stressors [21,22]. In pigs, music enrichment has been used to reduce aggression and increase play behavior [23,24], although the music used was originally created for humans. Two studies [25,26] found that both the harmonic structure and the structural features of music modulated emotional responses in pigs [25,26]. Therefore, it is important to use music that has been adapted in terms of harmonic and structural features to elicit positive responses in pigs.

Considering the above, this study aims to evaluate whether the use of adapted original music reduces aggressive behavior during regrouping, which is easily implemented environmental enrichment alternative [27].

## 2. Materials and Methods

### 2.1. Study Design and Population

The Ethics Committee in Animal Experimentation of the Universidad de Antioquia (CEEA) authorized all reported procedures on animals (Act No. 129 of 29 October 2019).

This experimental study was carried out on weaned pigs (n = 63) from a commercial production system (Hacienda La Montaña). The swine unit of the Hacienda La Montaña of the University of Antioquia is a full cycle production system, located in the northern region of the Department of Antioquia, Colombia, with an average number of 230 animals, and it develops its production at an altitude of 2350 m.a.s.l., an average temperature of 15 °C (22–7 °C), and a relative humidity of 72%.

Hacienda La Montaña is a commercial farm setting, and it is important to note that it was not a controlled trial. The piglets were housed in pens measuring 1.65 × 1.65 m, with a junior hopper-type feeder with a capacity of 12 kg feed, a pacifier drinker, a reinforced plastic floor with rectangular grids, and chains and plastic bottles serving as elements to bite, acting as forms of physical environmental enrichment Nutritional management included a commercial feed tailored to the specific production stage. The diet used comes from a factory that analyzes the raw materials utilized in the diet formulation to ensure the piglets’ health. Throughout the study period, all piglets were consistently provided with the same diet and type of concentrate, and environmental conditions (including daily records of temperature, relative humidity, and wind) remained unchanged. As the farm is in a tropical region, a natural light and dark cycle of 12 h each was maintained throughout the study. The piglets belonged to the Camborough-29 maternal/PIC-410 paternal genetic line.

### 2.2. Sample Size and Sampling

As this was a commercial-type farm in a real production environment, sample size calculation and sampling were estimated on a convenience basis, considering the number of pens available during the study period and the number of animals available for each pen. Based on farm management guidelines, a litter of piglets is assigned to a given pen. For this study, two pens/litters per treatment were utilized, each accommodating approximately 10 piglets. At the age of 30 days, the piglets were relocated to the weaner area, keeping piglets from the same litter together within one pen. It is a common practice for piglets not to undergo regrouping during this stage; instead, regrouping typically takes place when they transition to the grower and finisher housing.

Six random litters of 10-week-old piglets, averaging 25 kg each, were selected. Two litters were selected for each intervention: control group A (no musical stimulus), group B (AM/PM musical stimulus), and group C (AM musical stimulus). During the regrouping process, half of the pigs from one litter were exchanged with an equal number from another litter. The piglets relocated to the other litter were categorized as ‘intruders,’ while those that remained in their original litter were labeled as ‘residents.’ This arrangement resulted in approximately 20 piglets per group, divided into two pens, encompassing both residents and intruders.

To prevent control Group A (no musical stimulus) from listening to music and to ensure that intervention Group B (AM/PM musical stimulus) and Group C (AM musical stimulus) had no prior exposure to music, each trial was conducted with a 5-day difference. The experiment began with Group A in the nursery room, where they were not exposed to music. After the initial 5-day period, Group B was exposed to music (AM musical stimulus). Five days later, after the exposure of Group B (AM/PM musical stimulus), Group C (AM musical stimulus) was moved from the farrowing area to the nursery room, ensuring that the piglets had no prior exposure to music.

### 2.3. Interventions

The music used was of original composition (called Master 2 and 4). The selected music had a differentiating feature: it was adapted according to Zapata et al.’s 2023 study, as part of the results of QUIRON Pathobiology research group [26].

The term adapted refers to the compositions process of integrating and adjusting various spectral and temporal structures of music for the purpose of providing environmental enrichment by avoiding spectral and temporal structures of music that may generate fear or stress in pigs.

Each musical piece was guided by the analysis of its acoustic attributes, using musical Information Retrieval (MIR) techniques, and adjusting the acoustic parameters of the musical pieces according to the required spectro-temporal values necessary to induce calm in pigs. This analysis was conducted using Sonic Visualizer^®^ software version 4.2 (2020, Chris Cannam and Queen Mary, University of London), and it yielded numerical data that were crucial for tailoring the music to our research goals [26].

The adaptation process was carried out based on the following acoustic attributes:

Centroid: This parameter, which represents the center of mass of the sound spectrum and is related to sound brightness and timbre, was considered during the adaptation process.

Amplitude: The distance between the peak of the wave and its base, measured in decibels (dB), was another critical factor in the adaptation. Changes in amplitude were used to influence the intensity and volume of the music, impacting the behavior of the subjects.

Dissonance: Sensory dissonance, which measures the perceptual roughness of the sound, was assessed based on the roughness of spectral peaks. The adaptation process involved manipulating dissonance to affect the behavioral response.

High Frequency Content (HFC): The presence of high-frequency content in the sound spectrum was taken into consideration. Adjustments were made to this attribute to elicit specific behavioral responses in the subjects.

Zero Crossings Rate (ZCR): ZCR, which measures the number of times the signal crosses the zero axis, was used to differentiate between periodic and noisy sounds. This distinction was employed to influence the subjects’ reactions.

Pulse in Beats Per Minute (BPM): The tempo-rhythm of the music, quantified as BPM, was manipulated to affect the pace and rhythm of the subjects’ behavior.

Spectral Deviation: This parameter, indicating the standard frequency deviation around the spectral centroid, was adjusted to modify the frequencies in the spectrum and their deviation from the center of gravity, impacting the behavioral response of the subjects.

Instrumentation: The number of instruments played simultaneously was kept constant throughout the duration of each musical piece. This stability in instrumentation was maintained to ensure that any observed changes in behavior could be attributed to the variations in the other acoustic attributes.

Upon adapting each musical piece, they were compiled into a master. Each master consisted of five to six musical works, ensuring a Sound Pressure Level, measured using a sonometer, not exceeding 70 dB, to avoid stress in pigs [28,29] and a duration of 5 min in each musical piece, separated by periods of rest (silence) of 3 min, for a total duration of 1 h per master. The musical works were also adapted according to the auditory perceptive characteristics of pigs (pig hearing range, 40.5 Hz to 40 kHz, with a region of best sensitivity from 250 Hz to 16 kHz) [30], maintaining intra-work homogeneity in spectral or temporal acoustic parameters (e.g., low pulse content) over the entire duration [26]. Please note that these musical works are copyrighted.

The control group A (n = 20) had no intervention and remained under the usual management conditions of the farm. Group B (n = 21) was exposed to music from the time they were regrouped and for two continuous hours in the morning (8:00–10:00 a.m.) and in the afternoon (2:00–4:00 p.m.). Group C (n = 22) was exposed from the time of regrouping and for one continuous hour in the morning (8:00–9:00 a.m.) for 2 days. The variation in the number of individuals per group was due to farm management conditions. Regrouping of piglets occurred between 10:30 a.m. and 11:30 a.m. for all groups. The musical interventions were not simultaneous due to spatial conditions and availability of animals. 

The selection of music exposure duration was based on the perspective within our research group, where we view music as a potentially dose-dependent treatment. We hypothesized that the duration and timing of music exposure would yield different effects on the animals. Due to the limited existing literature on this specific topic, our research aimed to explore the optimal dosage for achieving the desired effects on the piglets’ behavior.

### 2.4. Data Collection

A 360° bullet camera (AHD camera, 2.0 mpx, Led array, bullet type, for outdoor use) was installed above the pens to record behavior continuously. After regrouping, 48 continuous hours (from 10 a.m. on day 1 to 48 h) were recorded using a continuous scanning observation method for each hour of recording, as suggested by Stukenborg et al., 2011 [5]. The video recordings were analyzed by three previously trained persons using BORIS software version 7.9.1. The videos were randomly assigned to each observer in a single-blind fashion, in which the observer did not know the purpose of the experiment and the group assignment.

The frequency (number of events) and duration (duration in seconds of each event) of the behaviors of Aggression (Negative Social Behavior; Aggression at the Feeding Trough) and Play (Social Play; Individual Play) were recorded. A new event was recorded if a pause of 8 s occurred during the development of the same behavior (“aggression” or “play”). The case definition for each behavior assessed in this study was established through an ethogram adapted from Casal et al. 2018 [31] and listed in Table 1.

### 2.5. Definition of Variables

The analysis included behavioral variables, intervention variables, and time variables. The behavioral variables were negative social or aggressive behavior; aggressive feeding behavior; social play; or individual play. The behavioral variables were measured in frequency (number of events) as a count variable and event duration (seconds) as a continuous variable. Intervention variables were music stimulus (control group A, treatment group B, treatment group C). Time variables were day (1, 3) measured as categorical variables.

### 2.6. Statistical Analyses

Normal distribution of all variables was checked graphically using histogram with a Gaussian distribution plot, scatter plots, and Shapiro–Wilk (W) test. Through a Tukey test, outliers (values more than 1.5 times the interquartile range from the quartiles, either below Q1 or above Q3) were removed from the data set, and variables with a W value < 0.9 were log transformed and checked for normality using Andersson Darling test (*p* > 0.05). Two separate analytical approaches were used to analyze the different types of outcome variables: (a) time series analysis for the variables of frequency and duration of behaviors versus intervention time and (b) Poisson regression model for count variables (number of behaviors). Confounders and interactions analysis was based on the presence of non-causal exposure-outcome associations. The variable day was included in the analysis as a potential confounding variable and stratified analysis (matching) was performed in which the effect of music on the different behaviors on days 1 and 3 was estimated separately.

Time series analysis was used to determine variations in frequency (discrete time series) and duration in seconds (continuous time series) of behaviors (aggressive or gambling) according to time and type of intervention. Repeated time measures were analyzed graphically, defining the behavioral event (separate plots for frequency or duration) and time, considering the behavioral observation hours in the morning (7–8/8–9/9–10/10–11) and the afternoon (11–12/12–1/1–2/2–3/3–4/4–5). Comparison tests were used to determine the differences between the musical intervention and the control (no musical stimulus) at a given time.

Poisson regression model was used to model events where outcomes are counted (number of aggressive/feeding/playing events) and assess the association between the musical intervention and the behavior of the weaned pigs during the regrouping procedure. For the model, an incidence rate ratio was used as a relative difference measure to compare the incidence rates of events occurring at any given point in time. For each behavior assessed, aggression or play, a separate Poisson regression model was run. The overall fit of the Poisson regression was assessed using Pearson’s goodness-of-fit and deviance tests. Residual analyses were also performed using Pearson and Anscombe. All analyses were performed using Stata^®^ statistical program (v. 17. College Station, TX, USA: Stata Corp LP).

## 3. Results

The behavioral variables (aggressive/feeding/playing) measured in seconds (duration) did not show a normal distribution of data; outliers were removed from the data set (as describe above) and were log transformed to be used in the time series analysis and as the offset variable for the Poisson models setting.

In Figure 1 of the time-course analysis, it is observed that for the aggressive behavior, on day one, after the first hour of regrouping (12–1 p.m.) and after the application of the musical stimulation, the groups that received the musical stimulation (Group B and Group C) had a significantly lower mean frequency of aggressive behavior than the group without musical stimulation (Control Group). The music stimulation groups (Group B and Group C) on day 1 not only displayed a reduced mean frequency of aggressive behavior but also a reduced duration of aggressive behavior, with a marked difference in the time periods of 2 p.m. and 4–5 p.m. on day 1 (Figure 1). Concerning the third day, it is observed that for the three groups, the aggressive behavior was still present 48 h after regrouping; however, the mean frequency of this behavior in the stimulated groups (Group B and Group C) was significantly lower at the times of 7–8 a.m. and 8–9 a.m., compared with the control group A. For Group B, a significantly greater difference is observed at 10–11 a.m. compared with the Group C and no musical stimulus group (A) (Figure 1). Regarding the mean duration of aggressive behavior, the musical stimulus groups had a significantly shorter duration, coinciding with the beginning of musical stimulus application at 8 a.m. (Figure 1).

For the behavioral measure of aggression during feeding, the groups with musical stimuli (Group B and C) show a significantly smaller difference in the mean frequency of presentation for days one and three compared with the group without musical stimuli (Group A). Regarding the mean duration on day three, this is lower in the groups receiving musical stimuli at 8–9 and 9–10, coinciding with the 8 a.m. stimulus application (Figure 2).

Regarding social play, although group B shows an increase in the mean frequency and duration of this behavior on day 1, this behavior is significantly reduced on day 3 for both groups with musical stimuli (Group B and C) compared with the Control group A without musical stimuli (Figure 3), and this is observed after the musical stimulus was reproduced for both groups at 8 a.m. For the individual play, on the third day and starting at 8 am, the mean frequency and duration of individual play were significantly lower compared with the group without musical stimulation (Group A), and as with social play, this reduction occurred after the musical stimulation (Figure 4).

The Poisson regression model with music stimulation as an exposure of interest and the number of behaviors as an outcome (Table 2) shows a significant association between music stimulation and behavior. In the groups tested, the rate of aggressive behavior was reduced by 0.67 (95% CI: 0.55–0.82) and 0.68 (95% CI: 0.55–0.83) for musical stimulations groups B and C, respectively, compared with the control group without musical stimulation when musical stimulation was provided on day 1 (Table 2). The rate of aggressive feeding behavior was reduced by 0.43 (95% CI: 0.23–0.81) and 0.34 (95% CI: 0.17–0.68) for groups B and C, respectively, compared with the control group. Regarding play, group C reduced the rate of occurrence of social play by 0.26 (95% CI: 0.13–0.52) and individual play by 0.48 (95% CI: 0.24–0.95) compared with the control group. 

Subsequently, when musical stimulation was provided on day 3 (Table 3), the rate of aggressive feeding behavior was reduced by 0.27 (95% CI: 0.15–0.49) and 0.25 (95% CI: 0.14–0.46) for groups B and C, respectively, compared with the control group. In relation to the play, group B and C reduced the rate of social play by 0.32 (95% CI: 0.23–0.45) and 0.22 (95% CI: 0.15–0.33), respectively, and individual play was reduced by 0.04 (95% CI: 0.005–0.30) and 0.50 (95% CI: 0.25–0.99), respectively, compared with the control group.

## 4. Discussion

The aim of the study was to evaluate whether the use of adapted original music reduces aggressive behavior during regrouping. To date, no studies have assessed pigs’ responses to adapted music for reducing aggressive behavior during regrouping, making these initial results significant. They demonstrate the effectiveness of adapted music as an environmental enrichment strategy in reducing aggression after regrouping. While other studies have used music, such as pop rock, during pig transport, these compositions were originally intended for humans and lacked specific behavioral goals for animals [23].

Kriengwatana et al. [32], who took a critical view of music for welfare research as auditory enrichment, suggest a different approach to the study of music for welfare, where music is used to address specific welfare goals by considering what animals hear in the music and selecting or creating musical compositions that test hypotheses about how music can influence animal behavior [32]. In accordance with the proposed approach, the music composed for this study was not only adapted to the pig’s preferences but was also composed with the intention of reducing aggressive behavior, demonstrating the usefulness of adapted music as an environmental enrichment strategy to reduce agonistic behavior during the regrouping process.

In the study by Stukenborg et al. [7], who measured aggressive behavior in piglets for 48 h after regrouping and reported that aggressive behavior in pigs has a circadian cycle, with aggressive interactions occurring primarily during the day, with more in the afternoon and fewer aggressive encounters at night [7]. Compared with our study, the circadian cycle, with an increased frequency of aggressive behavior from morning to afternoon, was observed in the group without music stimulation (control) on the first day of the experiment. However, this circadian cycle was broken in the music stimulation groups, especially on the first day, where after the first hour of regrouping, the music stimulation groups showed a decrease in aggressive behavior in the following hours.

The musical stimulation groups (Group B and C) not only reduced the frequency of aggressive behavior, but there was a positive effect in reducing the duration of these behaviors (Figure 1), which is positive for pig welfare, as increased frequency and duration of aggressive encounters have been associated with increased skin lesions. Studies have reported that dominant pigs have more lesions and higher frequencies and longer durations of aggressive encounters [7,33].

According to our results, it can be hypothesized that the use of adapted music during regrouping increases the resting behavior of pigs and therefore they spend less time fighting, which would have a positive impact on the welfare of the pigs. These findings align with a previous unpublished study conducted in the same farm, where we evaluated the effect of music on various behaviors, including resting behavior. In that study, we observed a reduction in the frequency of resting behavior, indicating the potential positive impact of music on pigs’ overall behavior. In another study with pregnant sows, it was found that those exposed to classical music were less active and therefore less vigilant and relate this situation as an indicator that the animals in the music stimulation group were more relaxed [34]. Contrasting, in our study the music created can be modified to modulate the behavior of the pig, compared with classical music composed for humans. As reported, the harmonic structure and structural features of music alters behavior responses in pigs [25,26].The intended music used during the experiment was adapted [26] to avoid spectral features that may cause distress in pigs, to reduce aggressive behavior.

Aggressive behavior was still present in the three groups. However, compared with the group without musical stimuli, the duration of aggressive behavior was significantly shorter in the groups with musical stimuli (Group B and C). This suggests that in the musical stimulus groups, aggressive encounters last for less time, which could be considered less intense, even though aggressive behavior may still occur 48 h after regrouping. Previous studies have found that aggressive encounters can occur up to 48 h after regrouping [35], suggesting that the hierarchy has not yet stabilized [7]. The Poisson regression model confirms that there was a reduced rate of aggressive behavior at the feeder compared with the population without musical stimulation. Therefore, the observation of aggressive encounters in our research is consistent with the literature, but the fact that aggressive behaviors lasted less time and had a reduced rate may indicate a stabilization of the pig’s hierarchy in the musical stimulation groups.

Regarding social and individual play, a reduction in play behavior was observed in the musical stimulation groups, especially on the third day. The results obtained indicate that the selected musical stimuli favored a calm behavior in the pigs, which explains the reduction in frequency and duration of play in the evaluated pigs compared with the non-musical group. These results are in contrast to those obtained by de Jonge et al. [24], who found that music facilitated play behavior in weaned piglets, although in their study the music acted as a conditioned cue, alerting them to the entrance of a playroom [24].

A strategy to reduce aggression during the regrouping process is to socialize the litter before weaning [36]. According to a study of producer perceptions of pig-to-pig aggression, 48% of producers do not consider the strategy of regrouping before weaning to be a practice and 49% do not implement it [19]. Therefore, the use of music during regrouping can be an easy strategy for producers to implement since it would only be necessary to play music at the time of regrouping and for 48 h afterwards to reduce the frequency and duration of aggressive behavior, and even, according our results, only playing music in the morning for 1 h (stimulus 2) would be enough to reduce aggressive behavior in pigs after regrouping. 

Regrouping is a stressful moment for the pigs. Not only does it destabilize the hierarchy, but it also causes injuries because of confrontations between pigs. Injuries that can occur in pigs cause losses for producers, mainly through carcass seizures, but can also reduce pigs’ physical and emotional well-being [37]. Therefore, this research is relevant to intensive production systems, as it provides a new strategy that is easy to implement to mitigate the negative effects of regrouping on the profitability of the producer and welfare of the pigs.

One of the limitations of the study is that it was conducted under the uncontrolled conditions of a commercial pig farm in Colombia. However, conducting this research within a commercially oriented farm enhances its practical relevance at the production level. Another limitation is the absence of another non-musical control group, such as white noise, due to farm design constraints. Including another control group with white noise would help ensure the specific impact of music as opposed to any auditory stimulus on pig behavior. Although, in another study conducted by the research group, pink noise, ambient noise, and music were evaluated, and it was observed that the responses were significant and different with music. Additionally, it is crucial to assess dietary factors, as they have the potential to adversely affect pig health and behavior. A significant concern is the possible bias related to aflatoxin contamination, which could significantly influence the research outcomes. Another limitation of the study was the farm’s design: the musical interventions were not simultaneous, and the study lacked repetitions. This discrepancy could have led to variations in microclimatic conditions. It is essential to recognize that the microclimate can introduce variations in piglet behavior, which should be controlled in future studies. 

Another limitation of the study was the farm’s design: the musical interventions were not simultaneous, and the study lacked repetitions. This discrepancy could have led to variations in microclimatic conditions. It is essential to recognize that the microclimate can introduce variations in piglet behavior, which should be controlled in future studies. Although we understand that the microclimate in each pen cannot be controlled, we have presented results under farm conditions that, when compared with a laboratory setting, may facilitate the translation of these results from trials conducted on an actual farm. We invite these findings to be verified under controlled conditions and are willing to collaborate on musical stimulus studies.

To the best of our knowledge, our findings represent an advance in understanding the use of adapted original music as an environmental enrichment for the reduction in aggressive behaviour in an intensive pig production system from a tropical region. However, there are several challenges involved in conducting animal behaviour studies linked to the production context of Colombia and this region. The first is obtaining large sample sizes with successful follow-up periods, which is due to the farm characteristics in the region: (1) small area of farms; (2) the small number of pigs in production; and (3) limited access to the farms. A second problem associated with conducting animal health/welfare studies in the production context of Colombia is the variations in farm management. Briefly, these include the absence of herd records, variations in nutritional/sanitary management, controlled personal entry for biosecurity with difficulties in accessing farms, and uncooperative farmers. Farmers may be reluctant to collaborate on studies that involve prolonged follow-up and routine sampling due to stress and economical losses in the pigs. For these reasons, we were not able to select herds at random; instead, we selected a farm that was representative of the population based on knowledge of the area and could facilitate sampling with continuous access to animals and herd information and the ability to control factors that could impede the gathering of data and their analysis.

Due to these limitations, sample size estimation was performed on a convenience basis, considering the number of pens available to obtain a reasonable minimum sample size in a commercial-type farm in a real production environment. The analysis included a final number of 63 animals distributed in three groups, in which significant differences in behaviour were found. The number of observations included in the analysis (assuming a significance level of 95%) corresponded to a power of 75% to detect differences in the duration of aggressive behaviour between the treatment groups and the control group. Despite a study population that could be considered small, the study population was well monitored, numerous useful measurements were made, and the variables were controlled using appropriate statistical methods.

Furthermore, in our study, we did not consider the possible habituation response that might occur when using musical stimuli during regrouping. We believed that within 48 h, a habituation response would not likely occur due to the short duration of the implementation. However, we encourage other researchers to explore this topic and assess the habituation effect concerning the use of music as environmental enrichment during regrouping.

In future studies, we plan to address the current limitations and continue researching how adapting the acoustic parameters affects the neurophysiological mechanisms, neuroendocrine responses, and behaviors in non-human animals. With this information, our goal is to design music enrichment programs with measurable objectives in productive systems. This will include production and economic parameters to evaluate the financial implications for producers using this music.

## 5. Conclusions

Our study demonstrates the efficacy of adapted music created as an environmental enrichment strategy in reducing aggressive behavior during the regrouping of pigs. The use of carefully tailored music led to a significant decrease in the frequency and duration of aggressive encounters, particularly on day one of regrouping. This disruption of the circadian cycle of aggression suggests a potential positive impact on pig welfare.

Additionally, while the reduced frequency of aggressive behavior is accompanied by a decrease in play behavior, the practicality and ease of implementing music during regrouping make it a valuable tool for improving pig welfare and minimizing the negative consequences of regrouping in intensive production systems. This evidence-based creative approach could guide future research in designing musical enrichment programs for non-human animals.

These findings offer a promising avenue for future research and practical application in the pig farming industry, providing a readily accessible means to enhance pig welfare while addressing the challenges associated with regrouping.

## Figures and Tables

**Figure 1 animals-13-03599-f001:**
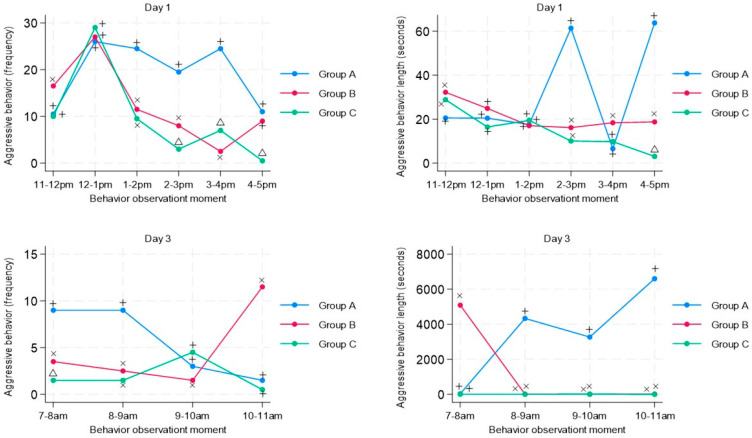
Time-series analysis of aggressive behavior in pigs (n = 63) subjected to musical stimuli at different moments during regrouping (day 1) and after regrouping (day 3). The figure shows the frequency (number of behaviors recorded in the pen) and duration (duration in seconds of each behavior) of aggressive behavior. A different letter indicates a significant difference (*t*-student test *p* < 0.05) in the frequency and duration of aggressive behavior of animals subjected to musical stimulus (Group B–Group C) compared with a control group without musical stimulus (Group A).

**Figure 2 animals-13-03599-f002:**
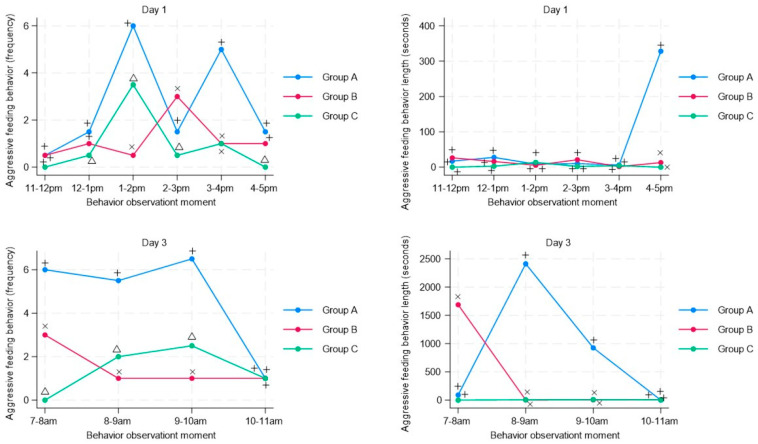
Time-series analysis of aggressive feeding behavior in pigs (n = 63) subjected to musical stimuli at different moments during regrouping (day 1) and after regrouping (day 3). The figure shows the frequency (number of behaviors recorded in the pen) and duration (duration in seconds of each behavior) of aggressive feeding behavior. A different letter indicates a significant difference (*t*-student test *p* < 0.05) in the frequency and duration of aggressive feeding behavior of animals subjected to musical stimulus (Group B–Group C) compared with a control group without musical stimulus (Group A).

**Figure 3 animals-13-03599-f003:**
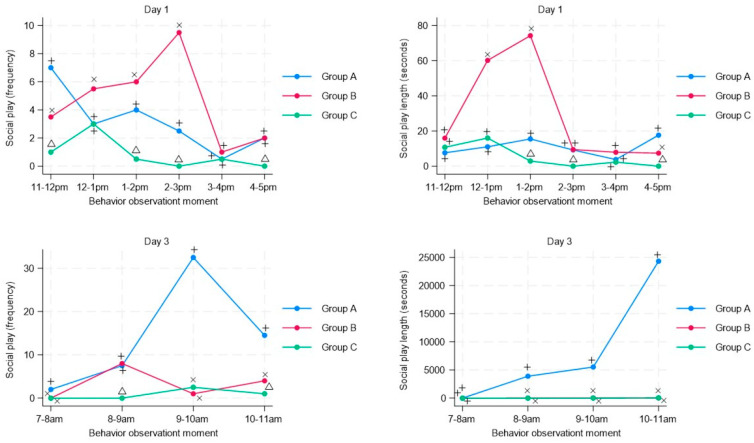
Time-series analysis of social play in pigs (n = 63) subjected to musical stimuli at different moments during regrouping (day 1) and after regrouping (day 3). The figure shows the frequency (number of behaviors recorded in the pen) and duration (duration in seconds of each behavior) of social play. A different letter indicates a significant difference (*t*-student test *p* < 0.05) in the frequency and duration of social play of animals subjected to musical stimulus (Group B–Group C) compared with a control group without musical stimulus (Group A).

**Figure 4 animals-13-03599-f004:**
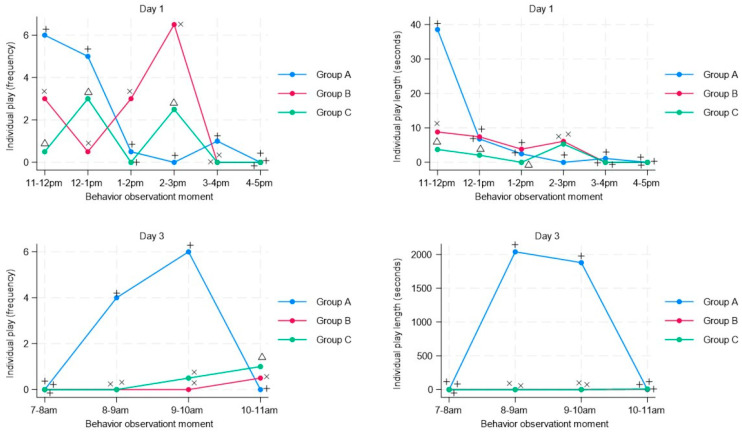
Time-series analysis of individual play in pigs (n = 63) subjected to musical stimuli at different moments during regrouping (day 1) and after regrouping (day 3). The figure shows the frequency (number of behaviors recorded in the pen) and duration (duration in seconds of each behavior) of individual play. A different letter indicates a significant difference (t-student test *p* < 0.05) in the frequency and duration of individual play of animals subjected to musical stimulus (Group B–Group C) compared with a control group without musical stimulus (Group A).

**Table 1 animals-13-03599-t001:** Ethogram: Case definition for behaviors (Aggressive behavior–Play) assessed in pigs during regrouping.

Behavior	Description
Social Negative Behavior	Ramming the opponent with the head (headbutting). Two pigs in head to head orientation, vigorously hitting each other, trying to bite ears, neck, or tail, and sometimes trying to lift the opponent by hitting with the muzzle under his body.
Aggression at the feeder	A pig eating or with its head inside the trough pushes other pigs, in which the aggressor pig is either head-butting or trying to bite the opponent, causing the other pig to move.
Social play	Pushing with other pigs, in which the pig uses either its head, neck, or shoulders to push the body of another pig with minimal to moderate force, occasionally resulting in the displacement of the other pig. It also includes gentle pushing, in which the pig uses its snout to gently touch a part of another pig’s body other than its nose, usually in rapid succession of movements.
Individual play	Running, jumping, hopping, prancing, turning around in the pen, energetic in forward movements in the pen. Often associated with agitation and occasional marginal/accidental contact with other piglets, such as pushing. Falling or dipping may also occur, where the pig falls from an upright position to a sitting or lying position on the floor of the pen.

Note: Adapted from Casal et al. (2018) [31].

**Table 2 animals-13-03599-t002:** Poisson regression (independent model for each behavior assessed) of the effect of musical stimulus on behavior in weaned pigs during regrouping (day 1) in an intensive production system.

Model/Behavior	Variable	Number of Recorded Behaviors				
		IRR	Std. Err.	*p*-Value	95% CI	
					LL	UL
Model 1 Aggressive Behavior	Intercept	19.33	1.26	<0.001	16.99	21.98
	Musical Stimuli			* 0.001		
	Control	Ref.	-	-	-	-
	Group B	0.67	0.069	<0.001	0.55	0.82
	Group C	0.68	0.070	<0.001	0.55	0.83
Model 2 Aggressive Feeding Behavior	Intercept	2.66	0.471	<0.001	1.88	3.77
	Musical Stimuli			* 0.001		
	Control	Ref.	-	-	-	-
	Group B	0.43	0.140	0.010	0.23	0.81
	Group C	0.34	0.120	0.002	0.17	0.68
Model 3 Social Play	Intercept	3.16	0.513	<0.001	2.30	4.35
	Musical Stimuli			* 0.001		
	Control	Ref.	-	-	-	-
	Group B	1.44	0.305	0.080	0.95	2.18
	Group C	0.26	0.093	<0.001	0.13	0.52
Model 4 Individual play	Intercept	2.08	0.416	<0.001	1.40	3.08
	Musical Stimuli			* 0.001		
	Control	Ref.	-	-	-	-
	Group B	1.04	0.291	0.889	0.60	1.80
	Group C	0.48	0.168	0.037	0.24	0.95

* *p* value for the entire categorical variable. IRR = Incidence Rate Ratio. LL = Lower Level. UL = Upper level.

**Table 3 animals-13-03599-t003:** Poisson regression (independent model for each behavior assessed) of the effect of musical stimulus on behavior in weaned pigs during regrouping (day 3) in an intensive production system.

Model/Behavior	Variable	Number of Recorded Behaviors				
		IRR	Std. Err.	*p*-Value	95% CI	
					LL	UL
Model 1 Aggressive Behavior	Intercept	6.00	0.772	<0.001	4.65	7.72
	Musical Stimuli			* 0.001		
	Control	Ref.	-	-	-	-
	Group B	0.75	0.147	0.145	0.50	1.10
	Group C	0.70	0.140	0.076	0.47	1.03
Model 2 Aggressive Feeding Behavior	Intercept	5.40	0.734	<0.001	4.13	7.05
	Musical Stimuli			* 0.001		
	Control	Ref.	-	-	-	-
	Group B	0.27	0.081	<0.001	0.15	0.49
	Group C	0.25	0.077	<0.001	0.14	0.46
Model 3 Social Play	Intercept	14.00	1.183	<0.001	11.86	16.52
	Musical Stimuli			* 0.001		
	Control	Ref.	-	-	-	-
	Group B	0.32	0.055	<0.001	0.23	0.45
	Group C	0.22	0.044	<0.001	0.15	0.33
Model 4 Individual Play	Intercept	2.40	0.489	<0.001	1.60	3.58
	Musical Stimuli			* 0.001		
	Control	Ref.	-	-	-	-
	Group B	0.04	0.042	0.002	0.005	0.30
	Group C	0.50	0.176	0.050	0.25	0.99

* *p* value for the entire categorical variable. IRR = Incidence Rate Ratio. LL = Lower Level. UL = Upper level.

## Data Availability

Data is unavailable due to privacy restrictions.

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
