# Peer review of "Adapted Original Music as an Environmental Enrichment in an Intensive Pig Production System Reduced Aggression in Weaned Pigs during Regrouping"

_animals, 2023, doi:10.3390/ani13233599_

Round 1

Reviewer 1 Report

Dear Authors,

Thank you for your manuscript detailing the use of music to reduce regrouping aggression in pigs.  The welfare of intensively farmed animals is of increasing importance, and any method that reports improve welfare, particularly at minimal cost, will be of interest to the readership of Animals.

General points:

For the most part, the paper is well written and coherent. The term “adapter/adjusted original music” is used three times before any attempt is made to explain what it was.  The explanation is then “adapted according to Zapata et al. (2013)” (line 107), but you don’t say adapted how or to do what.  In the abstract / simple summary I would start with something like “pig-specific music with a slow tempo designed to reduce aggression” (if that’s what it was) and then keep to the same terminology throughout.

Under the sub-section on study population, you must have an ethics statement, demonstrating that an animal ethics review was undertaken, and approval was granted for the study.  No animal work is publishable without such a declaration.

I am unclear as to the interventions for Groups B and C.  Was the first play of music at regrouping (i.e., 11 am on day 1)?  This is what is implied by line 118.  There also is no clear logic for 2 hours twice a day verses 1 hour once a day, the introduction would benefit from a justification to the selection of times / time points and also any evidence to suggest there might be a difference between the two treatments.  You also lack a non-music control (i.e., white noise) so it is hard to conclude that the results come from your specific bespoke music or just general auditory enrichment.  At the very least, this should be included as a limitation.

For the results, what is the reason you haven’t reported results for Day 2?  It would be interesting to see the results from during the intervention rather than just the carry-over effect on Day 3.

In the discussion (Line 314) you make a very bold statement about the “modulation of the emotional state of pigs”.  In reality, we have no idea about the emotional consciousness of any non-human species, and any implications about emotional state taken from behavioural observations require several leaps of assumption.  Personally, I would stick to reporting what you actually measured (and reported in the results).

The conclusion is very light.  This and the abstract might be the only bits of the paper that some people will read – so make it do justice to your work!

Specific points:

Line 57 – slightly meaningless figure.  What is the overall cost per annum to the industry (€0.37 makes it look like an inconsequential problem).

Line 61 – add citations after each point rather than grouping at the end of the sentence.

Line 62-66 – there is a lot of information here credited to a single citation.

Line 96 – I understand that you had a sample size imposed upon you, but providing a power calculation will provide the confidence that your sample size was big enough.

Line 110 – dB are a relative rather than absolute value.  Stating how it was calculated (baseline, weighting curve) gives clarity.

Line 112 – you need to cite the reference for pig hearing range, plus I’m sure your equipment didn’t respond over about 22 kHz at the most, so how do you justify “adapted to auditory perceptive characteristics of pigs”?

Line 123 – you say here regrouping was between 1030 and 1130, but at Line 178 you say it was at 11 am.  I know 11 am is between 1030 and 1130, but be consistent!

Line 151 – Why are you now calling Group A, B and C none, stimulus 1 and stimulus 2?  Again, be consistent.

These are all relatively minor points that are easily addressed, and I look forward to seeing this manuscript published.

Author Response

Please, also see the attachment 

Response to Reviewer 1 Comments

1. Summary

Thank you very much for taking the time to review this manuscript. Please find the detailed responses below and the corresponding corrections highlighted in the re-submitted files.

2. Point-by-point response to Comments and Suggestions for Authors

Comments 1: The term “adapter/adjusted original music” is used three times before any attempt is made to explain what it was.  The explanation is then “adapted according to Zapata et al. (2013)” (line 107), but you don’t say adapted how or to do what.  In the abstract / simple summary I would start with something like “pig-specific music with a slow tempo designed to reduce aggression” (if that’s what it was) and then keep to the same terminology throughout.

Response 1: We appreciate your constructive feedback regarding the terminology used to describe the music in our study. To enhance clarity and consistency throughout the paper, in the methods, we will provide a brief description of the term adapted. This will serve as a clear and concise reference for our readers, aligning with the intent of our music.

Hears the description that appears in the methods section, and you also can track the changes in line 148-181:

“The term adapted refers to the compositions created integrating and adjusting various spectral and temporal structural of music for the purpose of providing environmental enrichment by avoiding spectral and temporal structures of music, that may generate fear or stress in pigs.

Each musical piece was guided by the analysis of its acoustic attributes, using musical Information Retrieval (MIR) techniques, and adjusting the acoustic parameters of the musical pieces according to the required spectro-temporal values necessary to induce calm in pigs. This analysis was conducted using Sonic Visualizer® software (2018, Chris Cannam and Queen Mary, University of London), and it yielded numerical data that were crucial for tailoring the music to our research objectives[26].

The adaptation process was carried out, based on the acoustic attributes:

Centroid: This parameter, which represents the center of mass of the sound spectrum and is related to sound brightness and timbre, was considered during the adaptation process.

Amplitude: The distance between the peak of the wave and its base, measured in decibels (dB), was another critical factor in the adaptation. Changes in amplitude were used to influence the intensity and volume of the music, impacting the behavior of the subjects.

Dissonance: Sensory dissonance, which measures the perceptual roughness of the sound, was assessed based on the roughness of spectral peaks. The adaptation process involved manipulating dissonance to affect the behavioral response.

High Frequency Content (HFC): The presence of high-frequency content in the sound spectrum was taken into consideration. Adjustments were made to this attribute to elicit specific behavioral responses in the subjects.

Zero Crossings Rate (ZCR): ZCR, which measures the number of times the signal crosses the zero axis, was used to differentiate between periodic and noisy sounds. This distinction was employed to influence the subjects' reactions.

Pulse in Beats Per Minute (BPM): The tempo-rhythm of the music, quantified as BPM, was manipulated to affect the pace and rhythm of the subjects' behavior.

Spectral Deviation: This parameter, indicating the standard frequency deviation around the spectral centroid, was adjusted to modify the frequencies in the spectrum and their deviation from the center of gravity, impacting the behavioral response of the subjects.

Instrumentation: The number of instruments played simultaneously was kept constant throughout the duration of each musical piece. This stability in instrumentation was maintained to ensure that any observed changes in behavior could be attributed to the variations in the other acoustic attributes.”

In summary, the adaptation of each musical piece involved a thoughtful consideration of these acoustic attributes, which were assessed through quantitative computer analysis. Changes to these attributes were strategically made to design music that could effectively serve as environmental enrichment for non-human animals and improve their behavior in a desired manner. We will make sure to incorporate these details into our manuscript for clarity.

Comments 2: Under the sub-section on study population, you must have an ethics statement, demonstrating that an animal ethics review was undertaken, and approval was granted for the study.  No animal work is publishable without such a declaration.

Response 2: Thank you for pointing this out. You can locate the ethical statement in the paper on lines 104 and 105.

Comments 3: I am unclear as to the interventions for Groups B and C.  Was the first play of music at regrouping (i.e., 11 am on day 1)?  This is what is implied by line 118.  There also is no clear logic for 2 hours twice a day verses 1 hour once a day, the introduction would benefit from a justification to the selection of times / time points and also any evidence to suggest there might be a difference between the two treatments.  You also lack a non-music control (i.e., white noise) so it is hard to conclude that the results come from your specific bespoke music or just general auditory enrichment.  At the very least, this should be included as a limitation.

Response 3: We appreciate your feedback regarding the interventions for Groups B and C in our study. In the revised paper you may track the changes in line 191 – 205.

Interventions for Groups B and C:

You correctly observed that the first play of music for Groups B and C coincided with their regrouping at 11 am on day 1. We apologize for any ambiguity in our initial description. To clarify, for both Groups B and C, the first exposure to music occurred when the pigs were regrouped at 11 am on the first day. Subsequently, music sessions were conducted as follows:

Group B (n = 21): This group was exposed to music continuously for two hours in the morning (from 8:00 am to 10:00 am) and for two hours in the afternoon (from 2:00 pm to 4:00 pm) each day.

Group C (n = 22): Similar to Group B, this group's initial exposure to music was at the time of regrouping (11 am) on the first day. Following this, they received one continuous hour of music in the morning (from 8:00 am to 9:00 am) for two days.

Justification for Music Exposure Duration:

You correctly pointed out that our introduction lacks a clear justification for the selection of the music exposure duration and timing. We appreciate your feedback on this matter. In our research group, we view music as a potentially dose-dependent treatment. Hence, the experiment was designed to investigate whether varying the duration and timing of music exposure would yield different effects on the animals. Given the limited existing literature on this specific topic, our research aimed to explore the optimal dosage for achieving the desired effects on the piglets' behavior.

Absence of a Non-Music Control: (line 447 – 452 of the paper)

You raised a valid point regarding the absence of a non-music control group, such as exposure to white noise. We acknowledge this limitation. Due to infrastructure constraints, we were unable to create separate spaces to prevent animals from hearing both the music and white noise simultaneously. This limitation should indeed be included in our discussion as a potential confounding factor, and we appreciate your suggestion to highlight this.

We sincerely value your input, and your feedback will help us improve the clarity and transparency of our study.

Comments 4: For the results, what is the reason you haven’t reported results for Day 2?  It would be interesting to see the results from during the intervention rather than just the carry-over effect on Day 3.

Response 4 :

We would like to provide clarification on this matter.

Absence of Results for Day 2:

We acknowledge that we did not report results for Day 2 in our study, and we appreciate your interest in understanding the effects during the intervention itself rather than just the carry-over effect on Day 3.

The reason for not reporting results for Day 2 is the unforeseen technical difficulties that occurred on that day. Specifically, we encountered problems related to energy supply and a lack of recorder time, which affected our data collection process. These issues led to a partial loss of data for some hours in the morning, rendering the dataset incomplete and less reliable for a comprehensive analysis of that specific day.

Given the incompleteness of the Day 2 data, we made the decision to focus our analysis on the available and consistent data from Days 1 and 3 to ensure the reliability and accuracy of our findings. We believe that the analysis of Days 1 and 3 still provides valuable insights into the carry-over effects of our intervention.

Comments 5: In the discussion (Line 314) you make a very bold statement about the “modulation of the emotional state of pigs”.  In reality, we have no idea about the emotional consciousness of any non-human species, and any implications about emotional state taken from behavioural observations require several leaps of assumption.  Personally, I would stick to reporting what you actually measured (and reported in the results).

Response 5: We appreciate your constructive feedback on our manuscript.

Regarding your concern about our statement regarding the "modulation of the emotional state of pigs" in the discussion (Line 314), we acknowledge the uncertainty surrounding attributing emotional states to non-human species. In response to your suggestion, we will modify the discussion section to align with the principle of reporting only what has been directly measured and reported in the results, avoiding speculative assumptions about emotional states.

Comments 6: The conclusion is very light.  This and the abstract might be the only bits of the paper that some people will read – so make it do justice to your work!

Response 6: We have changed the conclusion to emphasize this point.

You may also track the changes highlighted in the re-submitted files (Line 496 - 509) and below:

“Our study demonstrates the efficacy of adapted music as an environmental enrichment strategy in reducing aggressive behavior during the regrouping of pigs. The use of carefully tailored music led to a significant decrease in the frequency and duration of aggressive encounters, particularly on day one of regrouping. This disruption of the circadian cycle of aggression suggests a potential positive impact on pig welfare.

Additionally, while the reduced frequency of aggressive behavior is accompanied by a decrease in play behavior, the practicality and ease of implementing music during regrouping make it a valuable tool for improving pig welfare and minimizing the negative consequences of regrouping in intensive production systems.”

Comments 7: Line 57 – slightly meaningless figure.  What is the overall cost per annum to the industry (€0.37 makes it look like an inconsequential problem).

Response 7: Thank you for highlighting this matter. We have addressed it, and you can review the revised passage below or highlighted in line 64-73:

" According to the findings, when tail biting lesions occur at an estimated frequency of approximately 10%, the related costs arising from this detrimental behavior can rise to €2.3 per pig at slaughter, equivalent to roughly 1.6% of the total carcass value [12]. However, the financial burdens linked to each pig experiencing tail biting lesions were significantly higher, as simulations indicated a range of €16 to €35 per affected pig. These costs were found to vary depending on the prevalence of tail biting lesions. It is suggested that the costs of tail biting lesions across Europe are approximately €2.0 (±€1.4) per finished pig [12].”

We hope this revision adequately addresses your comment

Comments 8: Line 62 add citations after each point rather than grouping at the end of the sentence.

Response 8: We have reviewed and revised the content as per your feedback, and the necessary changes have been implemented in the paper.

Comments 9:  line 62-66 there is a lot of information here credited to a single citation.

Response 9: We have reviewed and revised the content as per your feedback, and the necessary changes have been implemented in the paper in line 61 - 64

Comments 10: Line 96 I understand that you had a sample size imposed upon you, but providing a power calculation will provide the confidence that your sample size was big enough.

Response 10:

We have reviewed and revised the content as per your feedback. The changes are highlighted and appear in line 456-482 of the  paper and below:

“To the best of our knowledge, our findings represent an advance in understanding the use of adapted original music as an environmental enrichment for the reduction of aggressive behaviour in an intensive pig production system from tropical region. However, there are several challenges involved in conducting animal behaviour studies linked to the production context of Colombia and this region. The first is obtaining large sample sizes with successful follow-up periods, which is due to farm characteristics in the region: 1) small area by farms; 2) the small number of pigs in production; 3) limited access to the farms. A second problem associated with conducting animal health / welfare studies in the production context of Colombia is the variation of the farm management. Briefly, these include the absence of herd records, variation in nutritional/sanitary management, controlled personal entry for biosecurity with difficulties in accessing farms, and uncooperative farmers. Farmers may be reluctant to collaborate on studies that involve prolonged follow-up and routine sampling due stress and economical loses in the pigs. For these reasons, we were not able to select herds at random; instead, we selected a farm representative of the population based on knowledge of the area and facilitated sampling, continuous access to animals and herd information, and the ability to control factors that could impede the gathering of data and their analysis.

Due these limitations, sample size estimation was performed on a convenience basis, considering the number of pens available to get a reasonable minimum sample size under commercial-type farm in a real production environment. The analysis included final number of 63 animals distributed on three groups which significative differences in the behaviour were found. The number of observations included in the analysis (assuming a significance level of 95%) corresponded to a power of 75% to detect differences in the duration of aggressive behaviour between treatment group and control group. Despite the study population could be considered small, the study population was well monitored, numerous useful measurements were made, and the variables were controlled using appropriate statistical methods.”

Comments 11:  Line 110 dB are a relative rather than absolute value.  Stating how it was calculated (baseline, weighting curve) gives clarity.

Response 11: Thank you for your valuable comment concerning Line 110 and the use of decibels (dB) in our manuscript. Decibels are a relative measure of sound intensity, and we appreciate the opportunity to clarify how we handled this in our experiment.

In our study, we measured sound levels using a sonometer to ensure accuracy and consistency. These measurements were then adjusted based on established literature guidelines. We took into consideration that sound levels above 70 dB could potentially induce stress in the animals (line 182-184). Therefore, we carefully controlled the sound pressure level Sound (SPL) which can be described based on the decibel (dB) scale . This allowed us to maintain sound levels within a range that would not cause stress to the pigs.

Comments 12:  Line 112  - you need to cite the reference for pig hearing range, plus I’m sure your equipment didn’t respond over about 22 kHz at the most, so how do you justify “adapted to auditory perceptive characteristics of pigs”?

Response 12:

Thank you for your valuable comment. We have corrected the reference, and in our paper, we have provided a comprehensive explanation of the music design process.

Line ( 152 – 188)

Comments 13:  Line 123 – you say here regrouping was between 1030 and 1130, but at Line 178 you say it was at 11 am.  I know 11 am is between 1030 and 1130, but be consistent!

Response 13: We have reviewed and revised the content as per your feedback, and the necessary changes have been implemented in the paper. Line 197

Comments 14:  Line 151 – Why are you now calling Group A, B and C none, stimulus 1 and stimulus 2?  Again, be consistent.

Response 14: We have reviewed and revised the content as per your feedback, and the necessary changes have been implemented in the paper. Line 191-194

Reviewer 2 Report

Comments on the MS, with an ID ‘animals-2627427’ titled “Adapted original music as an environmental enrichment in an intensive pig production system reduced aggression in weaned pigs during regrouping” by Alvarez-Hernandez et al.

The article deals with one of the most important issues in pig production, namely weaning aggression, which escalates due to the sudden radical changes that piglets undergo at weaning (changes in the social and physical environment). The authors conducted an interesting study in which they used music (composed specifically for pigs to manipulate their behaviour) as an environmental enrichment to prevent excessive aggression in weaning piglets. In general, they found that the music was an effective means of preventing aggression, as the test group (with the music) fought less frequently, for shorter periods, and played more (with some exceptions). The study is undoubtedly very interesting and the approach interesting for breeders as it can be easily implied.

However, I am concerned about the methodology – many points are unclear (with the methodology described, it would be difficult to replicate the study), also the sample size and lack of replication do not allow for firm conclusions (e.g. "...strong evidence..." as you write in the abstract, line 36).

My specific comments can be found below:

L13-14: “… regrouping, which favors the presentation of aggressive behaviors to …”, this reads strangely, I don’t even understand it, so you mean something like ".... regrouping that triggers..."? Please rephrase.

L43: Reproduction, i.e. mates, is also considered a vital resource

L58-61: I agree with you that early socialisation is one of the most effective ways to prevent aggression at weaning, but I’m not sure about regrouping by sex, as many studies show that it is most important to keep whole original litters at weaning and not mix too many litters at once (search the literature). Rather than promoting hormonal/biotechnological interventions, I would suggest presenting some less invasive methods (i.e. changes in breeding technology itself) to prevent aggression. There are some known methods, e.g. imprinting, keeping whole litters at weaning, less litters mixed at weaning, providing more space, etc.

L87: “The piglets (weaned pigs)” use one of the two terms, not both.

L90: What physical environmental enrichment exactly?

Sample size and sampling:

Please provide more details about the study design and methodology – the comprehensive description of your methodology is, in my opinion, the main problem of your paper, which needs to be improved before MS is suitable for publication.  

1) In L99-102 you mentioned that there were two pens per treatment and 20 piglets per group and that a litter was assigned to a single pen. So you did not mix litters? As piglets establish a relatively stable hierarchy during lactation (suckling order), they are not expected to fight much after being moved to another pen.

2) You also speak of “regrouping”. What does that mean in the context of your study, because as far as I understand you did not regroup the animals, you just moved them from the farrowing pen to the weaning pen, right?

3) So the piglets were 10 weeks old, which is quite old for weaners (normally piglets are weaned at 4-6 weeks). So they were weaned at that age, which means the lactation period was 10 weeks, right? Or were they in another room/pen after weaning and before the experiment started?

4) Were the pens in a different room or in the same room? Perhaps provide a scheme of the study locations/pens/rooms.

5) Why are there no repetitions/replications? How can you rule out all random effects due to the specific conditions in an individual pen/room, e.g. different light conditions, air currents, microclimatic conditions, etc.? In my experience, the location of the pens/rooms can strongly influence the outbreaks of aggression (e.g. we once did an experiment in four pens in the same room and found that aggression was generally higher in one of the pens, of course we rule out this random error by replicating – so we assigned the treatment/control groups to the individual pens in a balanced manner). So, in order to draw a firm conclusion and exclude random effects/errors related to the specific conditions in a single room/pen, you should definitely do replicates/repetitions.

6) To repeat the study, you should also provide the audio file with the music used in the experiment.

L159: The number 1 at the beginning of the line is superfluous.

L164-167: I think it is not necessary to describe what is on the x-axis and what is on the y-axis if the reader knows what the variables (dependent and independent) are.

The results are generally difficult to comprehend, also because of the lack of information on methodology already mentioned.

L178: Please provide the details of the hours of “regrouping” in the M&M, not in the results.

L185: Between 2pm and 5pm the piglets were usually asleep, hence the low aggression in my opinion, right?

Please explain the Poisson regression models in a more intuitive way.

L327: You no longer need to refer to the tables in the discussion.

L346-347: Too firm conclusion.

L360-361: Have you also studied the resting behaviour? Because I can’t find it in the MS. The way it reads now, it implies that you have studied resting behaviour.

All the best!

I propose to edit the text for English.

Author Response

Please, also see the attachment

Response to Reviewer 2 Comments

1. Summary

Thank you very much for taking the time to review this manuscript. Please find the detailed responses below and the corresponding corrections highlighted in the re-submitted file.

2. Point-by-point response to Comments and Suggestions for Authors

Comments 1: L13-14: “… regrouping, which favors the presentation of aggressive behaviors to …”, this reads strangely, I don’t even understand it, so you mean something like ".... regrouping that triggers..."? Please rephrase.

Response 1: Thank you for pointing this out. We have made the changes highlighted in line 12-13, and below you may find the correction:

“In intensive swine production systems, managing the regrouping of pigs is a common practice, but it often leads to aggressive behaviors, which can harm the welfare of the animals”.

Comments 2: L43: Reproduction, i.e. mates, is also considered a vital resource

Response 2: We appreciate your feedback. In response, we conducted additional research to identify a suitable reference regarding mate resources and have incorporated it into our study, you may find it  in line 49- 50 of the paper and below:

Pigs are gregarious animals that establish hierarchical relationships to determine the order of access to resources such as food, water[3], mates and rest , [4].”

Comments 3: L58-61: I agree with you that early socialisation is one of the most effective ways to prevent aggression at weaning, but I’m not sure about regrouping by sex, as many studies show that it is most important to keep whole original litters at weaning and not mix too many litters at once (search the literature). Rather than promoting hormonal/biotechnological interventions, I would suggest presenting some less invasive methods (i.e. changes in breeding technology itself) to prevent aggression. There are some known methods, e.g. imprinting, keeping whole litters at weaning, less litters mixed at weaning, providing more space, etc.

Response 3: Thank you for your valuable suggestion. we have incorporated the consideration of space allowance and the practice of keeping whole litters together at weaning into our paper. In line 74 -82.

“Strategies such as early socialization during the first two weeks of life, regrouping pigs by sex at weaning [13], the use of synthetic maternal pheromones[14],  the use of tryptophan in the diet  [15], has been used to improve social skills in piglets and reduce aggressive interactions in future mixes, as well as the amount of time animals spend fighting. It has been demonstrated that decreasing the available area per individual, thereby increasing group density, heightens the frequency of agonistic interactions within the group. Therefore, increasing space and reducing stock densities may effectively mitigate piglet aggression[16]. However, it's essential to emphasize that a fundamental principle of the weaning process is to avoid excessive random mixing to prevent piglet aggression[17].” 

Comments 4: L87: “The piglets (weaned pigs)” use one of the two terms, not both.

Response 4 : Thank you for pointing this out. We have made the correction in line 132.

Comments 5: L90: What physical environmental enrichment exactly?

Response 5:  “Chains and plastic bottles serve as elements to bite, acting as forms of physical environmental enrichment”. This has been changed in the paper in line 115-16

Comments 6: In L99-102 you mentioned that there were two pens per treatment and 20 piglets per group and that a litter was assigned to a single pen. So you did not mix litters? As piglets establish a relatively stable hierarchy during lactation (suckling order), they are not expected to fight much after being moved to another pen

Response 6: Thank you for highlighting this matter. We have addressed it, and you can review the revised passage below and highlighted in Line 131 - 143:

“For this study, two pens/litters per treatment were utilized, each accommodating approximately 10 piglets. At the age of 30 days, the piglets were relocated to the weaner area, keeping piglets from the same litter together within one pen. Six random litters of 10-week-old piglets, averaging 25 kg each, were selected. Two litters were selected for each intervention: control group A (no musical stimulus), group B (AM/PM musical stimulus), and group C (AM musical stimulus). During the regrouping process, half of the pigs from one litter were exchanged with an equal number from another litter. The piglets relocated to the other litter were categorized as 'intruders,' while those that remained in their original litter were labeled as 'residents.' This arrangement resulted in approximately of 20 piglets per group, divided into two pens, encompassing both residents and intruders.”

Comments 7: You also speak of “regrouping”. What does that mean in the context of your study, because as far as I understand you did not regroup the animals, you just moved them from the farrowing pen to the weaning pen, right?

Response 7: Thank you for highlighting this matter. We have addressed it, and you can review the revised passage in  Line 138 - 141

Comments 8: So the piglets were 10 weeks old, which is quite old for weaners (normally piglets are weaned at 4-6 weeks). So they were weaned at that age, which means the lactation period was 10 weeks, right? Or were they in another room/pen after weaning and before the experiment started?

Response 8:  Thank you for your inquiry. The piglets were, indeed, weaned at the age of 30 days. Following weaning, they were relocated to the weaner housing area, where they remained. It is common practice for piglets not to undergo regrouping during this stage; instead, regrouping typically takes place when they transition to the grower and finisher housing

Comments 9:  Were the pens in a different room or in the same room? Perhaps provide a scheme of the study locations/pens/rooms

Response 9: We appreciate your suggestion and will incorporate it into the supplemental material the scheme of the farm and pens.

Comments 10:  Why are there no repetitions/replications? How can you rule out all random effects due to the specific conditions in an individual pen/room, e.g. different light conditions, air currents, microclimatic conditions, etc.? In my experience, the location of the pens/rooms can strongly influence the outbreaks of aggression (e.g. we once did an experiment in four pens in the same room and found that aggression was generally higher in one of the pens, of course we rule out this random error by replicating – so we assigned the treatment/control groups to the individual pens in a balanced manner). So, in order to draw a firm conclusion and exclude random effects/errors related to the specific conditions in a single room/pen, you should definitely do replicates/repetitions.

Response 10:  Thank you for your valuable comment. Due to biosecurity resources and constraints, it was not possible to conduct a replication. However, environmental and management conditions were consistent across all three groups. All tests were conducted in the same room with a natural light and dark cycle (12 hours light, 12 hours dark) (line 123-124)

Also in the methods section we included this information: (line 243-246)

Confounders and interactions analysis was based on the presence of non-causal exposure-outcome associations (Dohoo et al., 2002). The variable day was included in the analysis as a potential confounding variable and, stratified analysis (matching) was performed which, the effect of music on the different behaviors on days 1 and 3 was estimated separately

Comments 11:  To repeat the study, you should also provide the audio file with the music used in the experiment.

Response 11: The musical works used in the experiment are copyrighted. The property belongs to Universidad de Antioquia (UdeA) and the music should be requested through UdeA's technology management.

Comments 12:  L159: The number 1 at the beginning of the line is superfluous.

Response 12: Thank you for pointing this out. We made the correction.

Comments 13: L164-167: I think it is not necessary to describe what is on the x-axis and what is on the y-axis if the reader knows what the variables (dependent and independent) are.

Response 13: We have revised the content as per your feedback, and the necessary changes have been implemented in the paper.(line 270)

Comments 14:  L178: Please provide the details of the hours of “regrouping” in the M&M, not in the results.

Response 14: Thank you for highlighting this matter. We made the changes into the paper.(line 265)

Comments 15:  L185: Between 2pm and 5pm the piglets were usually asleep, hence the low aggression in my opinion, right?

Response 15: Since the farm is in a tropical region, a natural light and dark cycle of 12 hours each was maintained throughout the study. Therefore, piglets are typically asleep by 6 pm. This observation is consistent with findings from a previous study done by our research group.

Comments 16:  Please explain the Poisson regression models in a more intuitive way.

Response 16: Thank you for highlighting this matter. We made the changes into the paper in line 256-261 and below:

“Poisson regression model was used to model events where outcomes are counted (number of  aggressive / feeding / playing events) and assess the association between the musical intervention and the behavior of the weaned pigs during the regrouping procedure. For the model, an incidence rate ratio, was used as a relative difference measure to compare the incidence rates of events occurring at any given point in time. For each behavior assessed, aggression or play, a separate Poisson regression model was run. The overall fit of the Poisson regression was assessed using Pearsons goodness-of-fit and deviance tests. Residual analyses were also performed using Pearson and Anscombe. All analyses were performed using Stata® statistical program (v. 18. College Station, TX: Stata Corp LP)”.

Comments 17:  L346-347: Too firm conclusion.

Response 17: We expanded the conclusion you may find the changes in line 495

Comments 18:  L360-361: Have you also studied the resting behaviour? Because I can’t find it in the MS. The way it reads now, it implies that you have studied resting behaviour.

Response 18: thank you for your insight, we studied resting behavior in an unpublished study conducted in the same farm, where we evaluated the effect of music on various behaviors, including resting behavior. Line (396-401)

3. Response to Comments on the Quality of English Language

Point 1: I propose to edit the text for English.

Response 1: We have made several changes to the text in the hope of improving the overall quality of the English in the paper

5. Additional clarifications

Below you may find the scheme of the farm.

Reviewer 3 Report

The research, titled “Adapted original music as an environmental enrichment in an intensive pig production system reduced aggression in weaned pigs during regrouping.“ addresses an important and timely topic. I found the subject matter of the article fascinating and read the manuscript with great interest. The paper aligns well with the scope of the journal. However, I believe that in its current form, it has several shortcomings.

The aim of this study was to assess the effectiveness of using adapted original music as an environmental enrichment strategy to reduce aggressive behaviors during pig regrouping, a common practice in swine production known to cause stress and welfare issues. The study's main contributions include demonstrating that musical stimulation significantly reduces the frequency and duration of aggressive behaviors in weaned pigs during and after regrouping. This cost-effective and easy-to-implement enrichment strategy has the potential to improve pig behavior and welfare, providing valuable insights for swine producers seeking practical methods to enhance their production systems.

The paper addresses an important issue in swine production – the welfare of pigs during regrouping – and explores an innovative approach using music as an environmental enrichment strategy. While the study has several strengths, including its clear hypothesis and well-structured methodology, there are areas that could be improved. Firstly, the literature review could be more comprehensive, providing a deeper background on the effects of regrouping and existing strategies to mitigate aggression. Secondly, the economic implications of implementing this music-based enrichment should be discussed. Additionally, the study's limitations, such as the specific music selection and potential habituation effects, should be more explicitly addressed. Finally, the broader implications of the findings on pig welfare and production efficiency could be explored.

Specific comments:

I suggest rewriting the simple summary. According to the author's guidelines, this section should summarize and contextualize your paper within the existing literature in your field. It should be written without technical language or nonstandard acronyms, with the goal of conveying the meaning and importance of this research to non-experts.

I recommend rewriting the abstract and including more results and the significance of the obtained data.

Introduction:

I recommend expanding the introduction by providing a more comprehensive literature review on various aspects of animal welfare, including those related to regrouping and aggression in swine production. This will help set the stage for your study and provide a broader context for the music-based environmental enrichment approach. It's essential to establish the existing knowledge and gaps in the field of animal welfare before delving into your specific research focus. This will enhance the understanding of your readers and highlight the significance of your study within the larger context of swine production and animal well-being.

including references to studies that highlight welfare issues in various aspects of swine production can significantly enhance the comprehensiveness of your introduction. Here's a revised suggestion for expanding your introduction:

"In the context of swine production, ensuring the well-being of pigs is of paramount importance throughout their lifecycle. Challenges to animal welfare arise at various stages, including transportation, rearing conditions, nutrition, and social interactions. For instance, studies have demonstrated the stress and welfare problems that can occur during transportation (10.3390/ani10060945 and 10.3390/ani10122386), suboptimal rearing environments (https://doi.org/10.3390/ani9060383), and imbalances in nutrition (https://doi.org/10.1017/S0029665112000560). These welfare concerns manifest not only as physical health issues but also as behavioral problems, particularly during regrouping. Regrouping unfamiliar pigs often leads to aggressive behaviors, which not only affect the welfare of individual pigs but also pose management challenges in intensive swine production systems. While several strategies have been proposed to address these issues, such as modifying rearing conditions and nutrition, the present study investigates an innovative approach—using adjusted original composed music—as a cost-effective and easy-to-implement environmental enrichment strategy to reduce aggressive behaviors during regrouping."

By incorporating references to studies addressing welfare problems in various aspects of swine production, you provide a broader context for your research and underscore the significance of your chosen approach in improving animal welfare.

Methods:

t's crucial to include an ethical statement in your research paper, detailing the ethical considerations and procedures you followed while conducting the study. This statement ensures that your research adheres to ethical guidelines and principles, particularly regarding animal welfare and experimentation. Including this statement is a standard practice in scientific research to demonstrate transparency and ethical responsibility.

I suggest expanding the Methods section to provide a more detailed and comprehensive description of the procedures. This will enhance the clarity and replicability of your study. Consider including the following details:

To enhance the transparency and replicability of your research, I kindly suggest that you include a section detailing the methods employed for dietary analysis. This should encompass the techniques and procedures used to determine the composition of the diet. Furthermore, I recommend referencing a reputable source for these methods, such as the protocol outlined in 10.3390/ani13050797 and 10.3390/vetsci10090554.

The potential impact of aflatoxin levels in the feed, particularly in the corn used during the trial, on liver function and study outcomes is a valid concern. To address this issue and ensure the integrity of our study, we would like to confirm that rigorous quality control measures were implemented throughout the study. Specifically, the feed provided to the animals, including the corn, was regularly tested to ensure that aflatoxin levels remained well below established safety limits for animal consumption. This stringent monitoring was undertaken to mitigate any potential bias related to aflatoxin contamination, which could adversely affect liver health and consequently influence the study's results.

Please report a specific comment regarding the absence of this kind of bias of your study such as: "The diet provided in this study was carefully monitored to ensure that aflatoxin levels were well below the established safety limits for animal feed. This precautionary measure was taken to safeguard the animals' health and welfare. Aflatoxin contamination in animal feed can pose serious health risks, including impaired growth and liver damage (see, for example, 10.3390/toxins14070430). By maintaining feed quality within safe limits, we aimed to minimize any potential influence of aflatoxins on the study results."

Could you please clarify whether you conducted tests for normality and homogeneity on your data before proceeding with the statistical analysis? It's crucial to ensure that the assumptions underlying your chosen statistical methods are met. I recommend referring to the guidelines outlined in [proposed reference, e.g., 10.1080/1828051X.2020.1827990] for conducting such tests to maintain the rigor and reliability of your analysis.

Explain how the data were presented and whether any transformations or adjustments were made to the raw data. Clarify how outliers, if any, were handled in the analysis.

To facilitate transparency and future research, consider sharing the data and detailed methodology used in this study.

Results:

I kindly request that you improve the quality and readability of all the figures in your manuscript. The figures currently suffer from low resolution and size issues, which make it challenging for readers to interpret the visual data effectively. High-quality and appropriately sized figures are essential to enhance the overall clarity and impact of your research. We recommend revising all figures to ensure they meet the necessary standards for publication.

Discussion:

Starting the discussion section by reiterating the aim of the study can provide clarity and context for readers.

I kindly suggest expanding the discussion section of your paper to include practical applications and a thorough exploration of the study's limitations. This addition will enhance the overall value of your research and provide a more comprehensive understanding of its implications.

Providing insights into potential future research directions or practical applications for farmers based on the findings would enhance the paper's value.

I would like to encourage you to further stress the limitations of your study in the discussion section. This will help provide a more comprehensive context for your findings and assist readers, including fellow researchers and practitioners, in interpreting and applying your results effectively.

Conclusion:

I kindly suggest expanding the conclusions section of your paper to provide a more detailed and comprehensive report of the main findings. This will help readers better understand the significance of your research.

Please double-check the reference list to ensure that all references are included in the main text and vice versa.

Author Response

Please, also see the attachment

Response to Reviewer 3 Comments

1. Summary

Thank you very much for taking the time to review this manuscript. Please find the detailed responses below and the corresponding corrections highlighted in the re-submitted file.

2. Point-by-point response to Comments and Suggestions for Authors

Comments 1: I suggest rewriting the simple summary. According to the author's guidelines, this section should summarize and contextualize your paper within the existing literature in your field. It should be written without technical language or nonstandard acronyms, with the goal of conveying the meaning and importance of this research to non-experts.

Response 1: We appreciate your feedback. We have revised the simple summary with a focus on making it accessible to non-experts. You can review the changes in line 12-20 and below:

“In intensive swine production systems, managing the regrouping of pigs is a common practice, but it often leads to aggressive behaviors, which can harm the welfare of the animals. This study explores an approach that involves composing and producing music based on acoustic parameters established by our research group. The aim is to reduce aggressive behaviors in pigs, thereby enhancing pig welfare during regrouping Our findings indicate that this cost-effective and easy-to-implement strategy reduces aggressive behaviors in piglets during regrouping. This research offers valuable insights for producers, providing them with a practical way to enhance pig behavior and welfare while also contributing to the broader understanding of animal well-being in swine production systems.”

Comments 2: L43: I recommend rewriting the abstract and including more results and the significance of the obtained data.

Response 2:  We appreciate your feedback. You can review the changes in line 21-37 and below:

“In pig production systems, the practice of regrouping unfamiliar pigs is common, often leading to aggressive behavior. Although the effect of different musical genres composed for humans has been evaluated in pigs, to mitigate aggression, there have been few attempts to create music specifically for pigs. Here, we assess whether sensory stimulation through music, created by adapting the acoustic parameters in the sound mix, induces changes in the aggressive behaviors of pigs during regrouping. Six litters of piglets of 10-week-old were randomly selected and assigned to different treatments. The control group (Group A) received no intervention, while Group B was exposed to music for two continuous hours in the morning and afternoon from the time of regrouping. Group C received musical stimulation for one continuous hour in the morning following regrouping. A significant reduction in the frequency and duration of aggressive behaviors was observed in the groups that received musical stimulation during regrouping. Additionally, social and individual play behaviors showed a decrease in the musical stimulation groups. These findings provide evidence for the effectiveness of created music as a strategy in reducing aggressive behavior during pig regrouping, which can enhance the welfare of pigs and offer a practical solution for pig producers to minimize aggression and its associated negative impacts.”

Comments 3: I recommend expanding the introduction by providing a more comprehensive literature review on various aspects of animal welfare, including those related to regrouping and aggression in swine production. This will help set the stage for your study and provide a broader context for the music-based environmental enrichment approach. It's essential to establish the existing knowledge and gaps in the field of animal welfare before delving into your specific research focus. This will enhance the understanding of your readers and highlight the significance of your study within the larger context of swine production and animal well-being. including references to studies that highlight welfare issues in various aspects of swine production can significantly enhance the comprehensiveness of your introduction

Response 3: We appreciate your suggestion; we expanded the introduction. You can review the changes in line 41-48 and below:

“In swine production systems Maintaining good husbandry conditions and animal welfare is crucial. However, intensive swine production systems frequently induce stress due to factors like environmental conditions, transportation, and social interactions. Issues such as barren and overcrowded spaces without straw for bedding or rooting contribute to problems such as tail biting, impacting both the economic viability and welfare of pig production[1]. During transportation, concerns like fatigue, heat stress, and aggressive behavior are common welfare issues affecting pigs [2]. These issues not only impact physical health but also present behavioral challenges, as observed during regrouping”

Comments 4: L87: t's crucial to include an ethical statement in your research paper, detailing the ethical considerations and procedures you followed while conducting the study. This statement ensures that your research adheres to ethical guidelines and principles, particularly regarding animal welfare and experimentation. Including this statement is a standard practice in scientific research to demonstrate transparency and ethical responsibility.

Response 4 : Thank you for pointing this out. You can locate the ethical statement in the paper on lines 104 - 105.

Comments 5: To enhance the transparency and replicability of your research, I kindly suggest that you include a section detailing the methods employed for dietary analysis. This should encompass the techniques and procedures used to determine the composition of the diet. Furthermore, I recommend referencing a reputable source for these methods, such as the protocol outlined in 10.3390/ani13050797 and 10.3390/vetsci10090554.

Response 5:  Thank you for highlighting this matter, We have addressed it, and you can review the revised passage below:

This study was conducted on a commercial farm setting, and it is important to note that it was not a controlled trial. The commercial diet utilized in the study was analyzed at the factory. During the entire study duration, all three groups of animals were consistently provided with the same diet and type of concentrate.  The changes were made in line 117 - 121

Comments 6: The diet provided in this study was carefully monitored to ensure that aflatoxin levels were well below the established safety limits for animal feed. This precautionary measure was taken to safeguard the animals' health and welfare. Aflatoxin contamination in animal feed can pose serious health risks, including impaired growth and liver damage (see, for example, 10.3390/toxins14070430). By maintaining feed quality within safe limits, we aimed to minimize any potential influence of aflatoxins on the study results."

Response 6: (line 452-455)

Comments 7: Could you please clarify whether you conducted tests for normality and homogeneity on your data before proceeding with the statistical analysis? It's crucial to ensure that the assumptions underlying your chosen statistical methods are met. I recommend referring to the guidelines outlined in [proposed reference, e.g., 10.1080/1828051X.2020.1827990] for conducting such tests to maintain the rigor and reliability of your analysis.

Response 7:  

Thank you for highlighting this matter, We have addressed it in line 235-239

“Normal distribution of all variables was checked graphically using histogram with a Gaussian distribution plot, scatter plots, and Shapiro-Wilk (W) test. Through Tukey test, outliers (values more than 1.5 times the interquartile range from the quartiles, either below Q1 or above Q3) were removed from the data set, and variables with a W value <0.9 were log transformed and checked for normality using Andersson Darling test (P > 0.05).”

Comments 8: Explain how the data were presented and whether any transformations or adjustments were made to the raw data. Clarify how outliers, if any, were handled in the analysis.

Response 8:  

You may find the changes highlighted in line 266-269 and below:

The behavioral variables (aggressive / feeding / playing) measured in seconds (duration) did not show normal distribution of data, outliers were removed from the data set (as describe above) and were log transformed to be used in the time series analysis and as the offset variable for the Poisson models setting.

Comments 9:  To facilitate transparency and future research, consider sharing the data and detailed methodology used in this study

Response 9: We appreciate your suggestion, and we have made some changes to our methods, including a schematic diagram of the farm as a supplementary material. As for the data, we can provide it upon request

Comments 10:  I kindly request that you improve the quality and readability of all the figures in your manuscript. The figures currently suffer from low resolution and size issues, which make it challenging for readers to interpret the visual data effectively. High-quality and appropriately sized figures are essential to enhance the overall clarity and impact of your research. We recommend revising all figures to ensure they meet the necessary standards for publication.

Response 10:  Thank you for your recommendation. We improve the figures in the paper

Comments 11:  Starting the discussion section by reiterating the aim of the study can provide clarity and context for readers.

Response 11: Thank you for your recommendation. We followed your instructions and included the aim of the study and beginning of the discussion. line 364-370

Comments 12:  I kindly suggest expanding the discussion section of your paper to include practical applications and a thorough exploration of the study's limitations. Providing insights into potential future research directions or practical applications for farmers based on the findings would enhance the paper's value

Response1 2: Thank you for pointing this out. We made the correction. We add the limitations and future research into the discussion in line 444-494

Round 2

Reviewer 2 Report

Dear authors,

so all the tests were done in the same room. Was this also done at the same time? If not, describe the experimental design (timing) in more detail in M&M. If they were done at the same time, how can you be sure that there were no auditory influences between the tests, since all the animals were in the room, both the control animals and the animals that listened to music (once or twice a day). Just in case I am misinterpreting this, please describe the methodology in more detail, it means, in enough detail to allow anyone to repeat the study (even if you cannot provide a music sample, which already prevents your study from being completely repeated).

Considering the lack of repetitions, you can in no way be sure that the microclimatic conditions were the same in all pens (see again the example I gave). Please mention this in the M&M section to expose all the limitations of the study.

All the best!

Author Response

Please also see the attachment.

Response to Reviewer 2 Comments

1. Summary

Thank you very much for taking the time to review this manuscript. Please find the detailed responses below and the corresponding corrections highlighted in the re-submitted file.

2. Point-by-point response to Comments and Suggestions for Authors

Comments 1: so all the tests were done in the same room. Was this also done at the same time? If not, describe the experimental design (timing) in more detail in M&M. If they were done at the same time, how can you be sure that there were no auditory influences between the tests, since all the animals were in the room, both the control animals and the animals that listened to music (once or twice a day). Just in case I am misinterpreting this, please describe the methodology in more detail, it means, in enough detail to allow anyone to repeat the study (even if you cannot provide a music sample, which already prevents your study from being completely repeated).Considering the lack of repetitions, you can in no way be sure that the microclimatic conditions were the same in all pens (see again the example I gave). Please mention this in the M&M section to expose all the limitations of the study.

Response 1: Thank you for your valuable comment. We would like to explain that the musical interventions were not simultaneous due to spatial conditions and availability of animals. To prevent the control group from listening to music and to ensure that the intervention group had no prior exposure to music, the experiments were conducted as follows: The experiments began with the control group (Group A) in the nursery room. Five days later, Group B was exposed to music in the morning and afternoon in the same room. For group C, we had to wait ten days after the start of the experiment to allow time for the piglets to arrive from the farrowing area and to ensure that they had no prior exposure to music (Each test was carried out 5-days apart).

We also have made the changes into the paper, the experimental design in more detail in M&M sections is highlighted in line 144-151 and line 463-472, below you may find the correction:

Sample size and sampling:

Line 144 - 151

“To prevent control Group A (no musical stimulus) from listening to music and to ensure that intervention Group B (AM/PM musical stimulus) and Group C (AM musical stimulus) had no prior exposure to music, each trial was conducted with a 5-day difference. The experiment began with Group A in the nursery room, where they were not exposed to music. After the initial 5-day period, Group B was exposed to music (AM musical stimulus). Five days later, after the exposure of Group B (AM/PM musical stimulus), Group C (AM musical stimulus) was moved from the farrowing area to the nursery room, ensuring that the piglets had no prior exposure to music.”

Discussion:

Line 463 – 472

“Another limitation of the study was the farm's design: the musical interventions were not simultaneous, and the study lacked repetitions. This discrepancy could have led to variations in microclimatic conditions. It is essential to recognize that the microclimate can introduce variations in piglet behavior, which should be controlled in future studies.

 Although we understand that the microclimate in each pen cannot be controlled, we have presented results under farm conditions that, when compared to a laboratory setting, may facilitate translation of these results from trials conducted on an actual farm. We invite these findings to be verified under controlled conditions and are willing to collaborate on musical stimulus studies.”

Reviewer 3 Report

Good job!

Author Response

Thank you for all your comments that have helped us improve our paper.